# Internet Access and Use by Patients with Gynecologic Malignancies: A Cross-Sectional Study

**DOI:** 10.3390/cancers16091677

**Published:** 2024-04-26

**Authors:** Frederik Bach, David Engelhardt, Christoph A. Mallmann, Sina Tamir, Lars Schröder, Christian M. Domröse, Michael R. Mallmann

**Affiliations:** 1Faculty of Medicine, University of Cologne, Joseph-Stelzmann-Str. 20, 50931 Cologne, Germanydavid.engelhardt@posteo.de (D.E.);; 2Department of Obstetrics and Gynecology, University of Cologne, Kerpener Str. 34, 50931 Cologne, Germany; 3Center for Integrated Oncology Aachen Bonn Cologne Düsseldorf, Germany; 4Department of Surgery, Helios University Hospital Wuppertal, Heusnerstr. 40, 42283 Wuppertal, Germany; christoph.mallmann@helios-gesundheit.de; 5Department of Obstetrics and Gynecology, Hospital of the City of Cologne, Neufelder Str. 32, 51067 Cologne, Germany

**Keywords:** E-Health, gynecologic oncology, internet, cancer digitalization index

## Abstract

**Simple Summary:**

Internet resources, both technical and informal, have become important aspects of developing E-Health applications for cancer populations. Despite extrinsic and intrinsic patient factors applicable to all patient populations, there exist patient population-specific aspects that E-Health developers and physicians should be aware of when introducing E-Health offers to specific cancer patient populations. A better understanding of specific aspects of the digitalization of gynecologic cancer patients will help to develop specific E-Health offers for this patient population.

**Abstract:**

The influence of digitalization on information-seeking, decision-making properties of patients, therapy monitoring, and patient–physician interactions has and will change the global health sector tremendously. With this study, we add knowledge on the degree of digitalization, digital device availability, the use and availability of home and mobile internet access, and the willingness to use novel forms of patient–physician interactions in a group of gynecologic cancer patients. From July 2017 to March 2022, 150 women with a diagnosis of gynecologic malignancy at the University Hospital of Cologne participated in this questionnaire-based cohort study. Any one of three potential internet access devices (stationary computer, smartphone, or tablet) is owned by 94% of patients and the only patient intrinsic factor that is significantly associated with the property of any one of these internet access devices is age. The Internet is used daily or several times per week to assess information on their disease by 92.8%, 90.1% use the Internet for communicational purposes and 71.9% and 93.6% are willing to communicate with their treating physicians via E-Mail or even novel forms of communication, respectively. In conclusion, the predominant majority of gynecologic cancer patients can be reached by modern internet-based E-Health technologies.

## 1. Introduction

The influence of digitalization on information seeking, decision-making properties of patients, therapy monitoring, and patient–physician interactions has and will change the global health sector tremendously. The introduction of electronic health records among many health systems worldwide aims to reduce healthcare inequities and offers patients information on their personal care, health status, physician appointments, medications, and test results. In addition, more active aspects, such as communication portals between physicians, other healthcare providers and patients, exercise, nutrition, and psychological counseling delivered via E-Health will be introduced within these portals [1,2,3]. The beneficial effects of E-Health services and telemedicine were proven during the COVID-19 pandemic and during recent natural disasters, and particularly, the aspect of teleconsultation has gained so much attention that it has been implemented in routine care in many healthcare systems after the pandemic [4,5,6]. Other technical innovations such as wearable devices and biosensors to monitor patients’ well-being and symptoms need the internet as a prerequisite [7].

Consequently, patient access to adequate hardware, the internet, and patient acceptance of digital applications are the basis for the further adoption of digitalization strategies in health care. Despite the ongoing usage of the internet and internet applications, digitalization has been shown to be differently available and still depends on technical factors like access to a computer and broadband internet [8,9] and intrinsic “soft” patient factors such as age, gender, race, income, and education [8,9,10,11,12,13,14,15]. Although not all these aspects can be influenced, these factors are addressable, and studies show that free access to devices, the Internet, and technical assistance may eliminate disparities in portal use in disadvantaged populations [16]. 

Free access to devices, the internet, and technical assistance is of utmost importance as studies show that internet use and information seeking on the internet change treatment decisions, influence patients’ experienced degree of satisfaction with therapy decisions and psychosocial well-being, and improve shared decision-making [17,18,19].

With this study, we aimed to add knowledge on the degree of digitalization, digital device availability, the use and the availability of home and mobile internet access, and the use of information seeking on the internet of the group of gynecologic cancer patients. We aimed to identify subgroups of patients with inequity in the access and use of digitalization.

## 2. Materials and Methods

From July 2017 to March 2022, 688 women with a diagnosis of a gynecologic malignancy (ovarian cancer, cervical cancer, endometrial cancer, or vulvar cancer) diagnosed and treated at the University Hospital of Cologne, Germany, were invited to participate in this questionnaire-based study. We used a self-developed yet not validated questionnaire that included all aspects of the degree of digitalization and the internet use of the patients as described before (Appendix B) [9,20,21,22,23]. The questionnaire included demographic characteristics such as education, country of origin and relationship status, basic requirements for internet use such as computer ownership, location and type of computer use, frequency of computer use, sources of information, and the importance of different sources of information for breast cancer diagnosis. The use of the internet was assessed in the following four classic areas in which the internet can be used: information, communication, community, and e-commerce, and patient characteristics were associated with the data [9].

Digitalization indices that combine several factors of digitalization into one number are widely used to facilitate comparison of the degree of digitalization of people, societies, or countries. We calculated a cancer patient digitalization index as described before [9]. Briefly, 13 questions of our questionnaire that measure the degree of digitalization of patients were combined into one index as follows: 1. ownership of a computer (0 points = no, 1 point = yes); 2. assessment of one’s own computer experience (0 points = no computer experience, 1 point = low computer experience, 2 points = good computer experience, 3 points = very good computer experience); 3. internet access at home (0 points = no, 1 point = yes); 4. usage of the internet (0 points = no, 1 point = indirectly via friends/family, 2 points = self-usage); 5. frequency of internet usage (0 points ≤ 1×/month, 1 point = several times/month, 2 points = several times/week, 3 points = daily); 6. usage of the internet as an information source (0 points = no, 1 point = yes); 7. frequency of the internet usage as an information source (0 points ≤ 1×/month, 1 point = several times/month, 2 points = several times/week, 3 points = daily); 8. usage of the internet for communication (0 points = no, 1 point = yes); 9. usage of the internet for communication (1 point = indirectly via friends/family, 2 points = self-usage); 10. usage of the internet for shopping (0 points = no, 1 point = indirectly via friends/family, 2 points = self-usage); 11. willingness to contact a physician via E-Mail (0 points = no, 1 point = maybe, 2 points = yes); 12. phone usage (0 points = stationary phone, 1 point = cell phone without internet, 2 points = smartphone); and 13. willingness to contact a physician via novel communication channels (0 points = no, 1 point = maybe, 2 points = yes). Each point was multiplied by 4, so a maximum of 100 points per patient could be achieved.

Statistical analysis was performed using SPSS 27.0 statistical software (IBM Corporation, Armonk, NY, USA). Ordinal scale data such as frequency of internet use, computer experience, or education were analyzed using the Spearman correlation coefficient, where appropriate. Nominal scale data such as computer property, cancer entity, internet access, or internet use were analyzed using Chi-square tests. A *p*-value of <0.05 was considered significant. Data were visualized using Microsoft Excel 2023 and CorelDRAW Graphics Suite 2020 (Corel Corporation, Ottawa, ON, Canada). This study was positively evaluated by the ethics committees of the University of Cologne (Ethics vote 17-146, 15 May 2017).

## 3. Results

A total of 688 patients with a gynecologic malignancy were approached at the gynecologic cancer center of the University Hospital of Cologne in the study period. Of these, 150 patients participated in this study (Table 1).

### 3.1. Digital Device Availability

Access to the internet as well as the availability of technical equipment for access, stationary at the beginning of the internet and with an increasing proportion via mobile equipment, represent the prerequisite for digital equity among different patient populations. Of the patients, 91.1% (133/146) owned a computer, and 76.8% (113/147) rated their computer knowledge as either good or very good (Figure 1, Appendix A).

Computer ownership was significantly associated with age (*p* = 0.009 for age </>60 years), whereas educational status, origin of the patient, household size, and cancer entity did not significantly contribute to computer property. Ownership of a smartphone was significantly associated with age (*p* = 0.009 for age </>60 years), educational status (*p* = 0.01), and origin of the patient (*p* = 0.032), yet it was not associated with household size or cancer entity. Interestingly, ownership of a smartphone was significantly higher in the patients who also own a computer (*p* < 0.001). Ownership of a tablet was significantly higher in the patients with a higher educational status (*p* = 0.003) and in the patients who also own a computer (*p* = 0.037).

As the exclusive use of the internet via a stationary computer has been abandoned widely considering the increasing amount of people that use smartphones with internet access or tablets instead of a stationary computer device to access the internet, we also assessed the availability of one of these three internet access devices of the patients. Overall, 94% of the patients (141/150) own either one of these three internet access devices. The only intrinsic patient factor that was significantly associated with the ownership of any one of these Internet access devices was age (*p* = 0.05 for age </>60 years), whereas availability of at least one of these internet access devices was not significantly different among different household sizes, educational status, origin of the patient, or tumor entities.

### 3.2. Digital Device Use

As outlined above, with 94% of the patients owning an internet access device, the requirement for access to the internet and the use of E-Health offers can be considered to be fulfilled. Furthermore, the use of computers and the internet might be interesting. Consequently, we next assessed the use of computers and the internet. Overall, 76.9% (113/147) of the patients identified their computer experience as good or very good. High computer experience was significantly associated with age (*p* < 0.001), educational status (*p* < 0.001), and the property of any one of the potential internet access devices (*p* < 0.001), yet it was not different between patients of different cancer entities or origins.

### 3.3. Internet Access

In addition to the ownership of an internet access device, access to the internet is a prerequisite for E-Health use. Overall, 95.8% (138/148) of the participants had internet access at home. The patients <59 years of age showed nearly full coverage of internet access at home, yet internet coverage was also high in the patients between 60 and 69 years (93% (40/43)) and even in the patients older than 70 years (90.5% (19/21)). Consequently, the differences in internet access between the different age groups were not statistically significant. In addition, internet access was high throughout different levels of education (primary school education: 90% (18/20); middle school education: 95.5% (42/44); and high school education: 97.5% (77/79)), different household sizes (living alone: 91.8% (45/49); household size ≥ two people: 97.8% (89/91), and different sub-entities of gynecologic cancer (ovarian cancer patients: 95.7% (44/46); cervical cancer patients: 100% (38/38); vulvar cancer patients: 93.9% (31/33); and endometrial cancer patients: 92.6% (25/27)), and there were no significant differences in internet access between the different groups. Factors that were significantly associated with internet access included the availability of and experience with a computer (each *p* < 0.001).

The majority (91.7% (133/145)) reported using the internet by themselves, whereas 2.1% (3/145) used the internet indirectly via friends or family members, and 6.2% (9/145) reported not using the internet. Self-usage of the internet was significantly associated with age (*p* = 0.024), educational background (*p* = 0.013), the availability of a computer (*p* < 0.001), own computer experience (*p* < 0.001), and the availability of the internet at home (*p* < 0.001), yet it was not associated with cancer entity, household size, or origin of the patient.

Overall, 77% (107/139) of the patients reported using the internet daily, whereas 15.8% (22/139) used the internet several times per week, 3.6% (5/139) used the internet several times per month, and 3.6% used the internet less than once monthly. Frequent internet use was significantly associated with younger age (*p* = 0.002), educational background (*p* < 0.001), the availability of a computer (*p* < 0.001), own computer experience (*p* < 0.001), the availability of the internet at home (*p* < 0.001), and the self-usage of the Internet (*p* < 0.001), yet not with cancer entity, household size, or origin of the patient.

### 3.4. Use of the Internet as a Source of Information

The internet has been adopted widely by patients as a source of information on their disease, but this use differs tremendously among disease entities and patient groups. Consequently, we assessed the use of the internet as a source of information for patients with gynecologic malignancies. Interestingly, 98.6% (139/141) of the patients with gynecologic cancer use the internet as a source of information and among these patients, the vast majority of 92.8% (129/139) use the internet daily or several times per week to assess information on their disease. Most of the patients search for information regarding cancer therapy (56.5% (70/124)), cancer research (53.7% (66/124)), general information (49.2% (61/124)), nutritional aspects (45.2% (56/124)), cancer specialists (37.1% (46/124)), and alternative therapies (21.8% (27/124)). Consequently, the websites of cancer societies (59.1% (65/110)), cancer aid (47.3% (52/110)), the gynecologic hospital (44.5% (49/110)) or the Gyneco-oncologist (15.5% (17/110)), and oncologic journals (29.1% (32/110)) are the most visited websites.

Despite the vast amount of information on cancer that is available nowadays on the internet, only 0.7% of the patients (1/149) stated that there was sufficient information by solely using the internet with no need for additional information from the respective physician, and only 16.1% (24/149) regarded the information obtained from the internet to be sufficient enough that they only needed validation of that information by their physician. The patients regarded the information obtained from the internet as insufficient with the need for additional information from their physician (14.8% (22/149)), or vice versa, the information obtained by their physician as insufficient with the need for additional information from the internet (7.4% (11/149)). The patients who stated that they obtained sufficient information from their physician mainly (49.7% (74/149)) used the internet for additional information purposes, and a minority stated no additional informational needs (11.4% (17/149)).

When asked for potential reasons not to use the internet for information on cancer, the patients stated either to be afraid to obtain false (30.4% (21/69)) or inaccurate (42.0% (29/69)) information or to obtain no sufficient information regarding their cancer subtype (11.6% (8/68)).

### 3.5. Use of the Internet as a Source of Information, Physician–Patient Interactions, and Therapy Decision-Making

Information seeking on the internet had a huge influence on physician–patient interactions as 52.5% (63/120) of the patients stated that they had discussed findings from the internet with their physician in the past, 7.8% (9/115) of the patients had found novel treatments on the internet, and 22.8% (26/114) of the patients had even found information on the internet that changed their cancer treatment. 

Consequently, only 7.7% (11/143) of the patients wished for their physicians to decide their cancer therapy alone. The majority prioritized either a decision of their physician influenced by their preferences (44.1% (63/143)) or shared decision-making (35.0% (50/143)). Only a minority wanted to decide either completely alone (0%) or influenced by their physicians’ recommendations (13.3% (19/143)).

### 3.6. Use of the Internet for Communication Purposes

Communication via mobile apps and the internet has changed the telephone landscape in the last 20 years tremendously. The majority (90.1% (128/142)) of the patients used the internet for communication purposes. Among those, 97.7% (125/128) used the internet by themselves, 1.6% (2/128) indirectly via their friends, and 0.7% (2/128) indirectly via their family members. Nevertheless, the majority still used a telephone to contact the oncologic outpatient clinic (75.7% (109/144)), with only 36.1% (52/144) and 1.4% (2/144) using E-Mail or Instant messaging communication channels. This might be influenced by the communication channels offered to the patients, as the patients’ willingness to communicate with their treating physicians via E-Mail (71.9% (87/121)), even with new and not yet established forms of communication channels (93.6% (118/126), was comparable to their willingness to use the traditional form via the telephone (90% (108/120)).

### 3.7. Digitalization Index

The cancer patient digitalization index includes the main aspects of digitalization and identifies groups of patients with a lack of digitalization, as can be seen in Figure 2.

Overall, 59.3% (89/150) of the patients showed a degree of digitalization of 70 or more (Figure 2). Higher scores in the digitalization index were significantly associated with younger age and higher education, whereas the cancer entity in the spectrum of gynecologic malignancies was not associated with a higher or lower digitalization status.

## 4. Discussion

Our study shows the high availability of electronic devices that allow access to the internet and consequently to internet-based informational tools and internet-based communicational tools for patients as compared with the status only some years ago [9].

In contrast to data on breast cancer patients, where not only age but also low educational background and household size have been associated with inequity in the access to computers and the internet and the use of the internet, in gynecologic cancer patients, only age was associated with diminished access to computers and the internet [9]. Although educational factors and age are still associated with stationary computer availability, most likely, easier access to smartphones and tablets has somehow reduced the disparity in terms of internet access among different socioeconomic groups. Nevertheless, age remains associated with reduced internet access in our patient cohort of gynecologic cancer patients, which is in line with other reports in the general population [15,24]. As a high proportion of gynecologic cancer patients falls in the age group above 60, these aspects must be incorporated into clinical trials and E-Health projects aimed at patients of this cancer entity.

In line with other studies on this topic, we identify the internet as an important source of information for gynecologic cancer patients [9,13]. The vast majority of the patients obtain information regarding therapies, discuss this information with their physician, and state that this information has changed their treatment. Consequently, the aspect of information gathering remains an important aspect of the internet for gynecologic cancer patients and enables them to be active healthcare users [25,26].

Over time, the internet has changed its function in general society, from an information resource at the beginning to a major social and communication platform at present. Further transformations that include virtual reality, artificial intelligence, and gamification aspects are likely to occur, and similar changes are likely to occur in the digital health system as well. In contrast to the past, an important aspect of the digitalization of health care has been the quantitative explosion of information regarding any aspect of diagnosis, and patient care and treatment options, patient portals, patient-reported outcome measures, and bidirectional communication portals will be only a question of time [2].

Whereas at the beginning of research in this area, there were mixed effects on patients’ healthcare quality [27], the positive impact of novel digital solutions to improve cancer care and the positive effects on quality of life and psychological outcomes have nowadays been shown in a multitude of different settings and studies [28].

As with all novel technologies, barriers exist that exclude certain groups from these developments [2,8,27]. These barriers might be technical or emotional [29]. Technical barriers include the lack of the availability of a computer or broadband access at home [30,31]. In addition to technical barriers, social, emotional, and financial barriers such as low household income exist [29]. Turner and colleagues identified intrinsic “soft” patient factors such as lack of experience, privacy concerns, and preferences of speaking directly to a healthcare provider as barriers to patient portal access [2]. Torrent-Sellens identified that healthcare usage, larger family size, and younger age were predictive of E-Health usage [29]. Among 28,942 patients eligible for electronic health record-linked portals, Griffin and colleagues reported that 35% of patients never accessed the portal [8]. Gender, racial minority, rural dwelling, and unemployment represented similar “soft” factors that were associated with non-use. Similar to our study, these studies clearly identify specific subgroups of patients with reduced access to novel healthcare offers. Interestingly, throughout most of the studies on this topic, socioeconomic status, low education, and older age again and again have been associated with digital health inequity [14].

An appreciation of the inequity in these specific subgroups is absolutely mandatory to design specific programs that introduce E-Health offers to these subgroups [32]. Rivière and colleagues reported on the effectiveness of a telemonitoring platform in cancer patients older than 70 years that might improve health care in this patient group [33]. Grosman and colleagues showed that free access to devices, the internet, and technical assistance may eliminate inequities in portal use in disadvantaged populations [16]. Bertera reported a study with older patients from a low socioeconomic background that were successfully skilled in the use of E-Health offers and the internet [34]. Despite the initial cost argument, over the long term, these interventions fostering telemedicine might be a cost-effective strategy to reduce costs in other places of the health system [35,36,37].

The limitations of our study are the relatively low number of patients in specific subgroups, especially in the patient group under 40 years. Consequently, we cannot discount that the results in the age group under 40 might be different from the results seen in the older age groups. Yet, as the usage of the internet increases with age in our cohort, we believe that the results are not influenced strongly by this fact. Another and probably more serious limitation is the risk of self-reporting. As we used a self-reported questionnaire, digital skills could not be assessed objectively and might have been reported as being too high or too low. In addition, a selection bias of patients that did or did not answer our questionnaire cannot be ruled out as only 150 patients were willing to participate in our study out of 688 patients that were approached. Nevertheless, as the questionnaire was a paper-based questionnaire, no specific technical skills or electronic devices were necessary to participate. As the questionnaire was in the German language, we cannot rule out that patients from an origin other than Germany might have been underrepresented in our study. This patient population might show another extent of digitalization than native German patients and might be approached specifically in future studies on this topic.

Our study contributes to the knowledge of the digitalization status of cancer patients with a specific focus on female patients with gynecologic malignancy. Our study identifies patients older than 60 as the patient subgroup that nowadays would have lower access probability to novel E-Health offers that are regularly introduced by cancer societies and pharmaceutical companies. In line with studies in other patient entities, the internet has not replaced the physician as the main informational resource for patients with gynecologic cancer. However, physicians are increasingly confronted with patients who actively seek information on alternative therapies and other specialists in the field. This potential benefit with regard to patient empowerment and shared decision-making might improve clinical care yet also represents a risk for false and misleading information, and the patients in our study were already aware of this fact. 

## 5. Conclusions

The predominant majority of gynecologic cancer patients can be reached by modern internet-based E-Health technologies. As there remains a proportion of patients, especially in the lower-educated and older population with lower digitalization status, we propose that the digitalization status of each patient should be obtained at the beginning of therapy to optimize digital patient–physician interaction and E-Health offers.

## Figures and Tables

**Figure 1 cancers-16-01677-f001:**
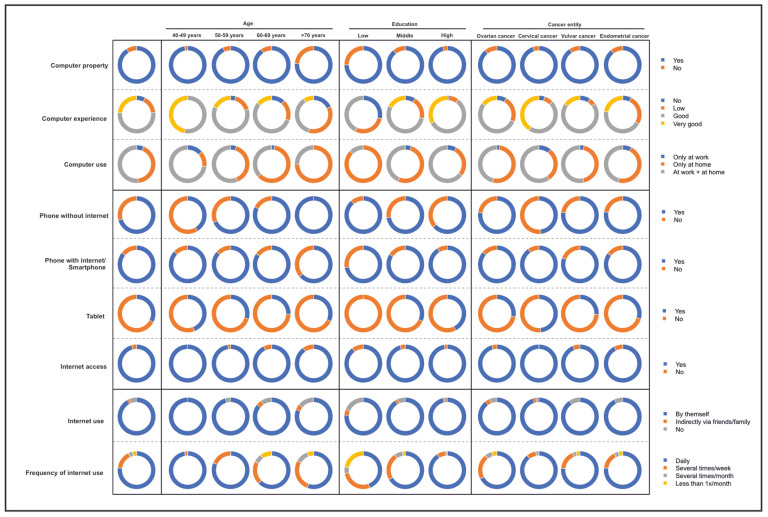
Availability of technical and informal resources in terms of devices and competence in use, differentiated by age, level of education, and cancer entity.

**Figure 2 cancers-16-01677-f002:**
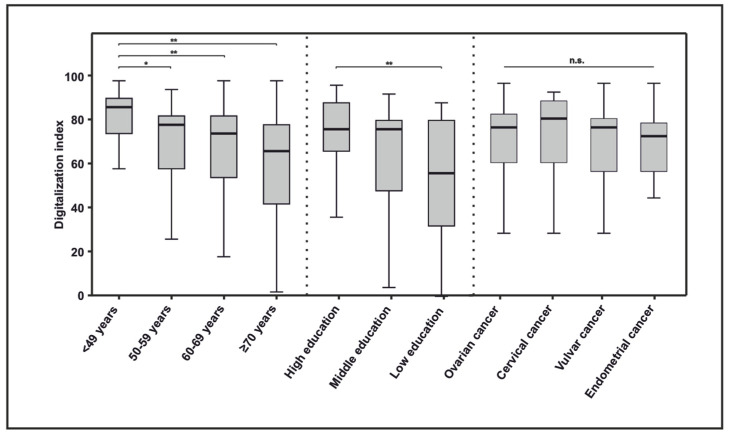
Digitalization index according to age, level of education, and cancer entity. (**: *p* < 0.001; *: *p* < 0.05; n.s.: not statistically significant).

**Table 1 cancers-16-01677-t001:** Characteristics of 150 participating patients with gynecologic malignancies.

Parameters		N or Mean (Standard Deviation)
Age (spread)		56.2 years (20–82)
	Age ≤ 49 years	39/150
	Age 50–59 years	45/150
	Age 60–69 years	43/150
	Age ≥ 70 years	23/150
Cancer entity	Ovarian cancer	48/150
	Cervical cancer	39/150
	Vulvar cancer	35/150
	Endometrial cancer	28/150
Origin	Germany	120/147
	Other	27/147
Education	No degree/low educational attainment	21/148
	Middle educational attainment	47/148
	High school/College degree	80/148
Household size	Living alone	49/145
	Household size ≥ two persons	96/145

## Data Availability

The original contributions presented in this study are included in this article. Further inquiries can be directed to the corresponding author.

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
