# Peer review of "Internet Access and Use by Patients with Gynecologic Malignancies: A Cross-Sectional Study"

_cancers, 2024, doi:10.3390/cancers16091677_

Round 1
Reviewer 1 Report
Comments and Suggestions for Authors
I have revised the manuscript titled “Internet access and use of patients with Gynecologic Malignancies: a cohort study”.
The Study is focused on the interesting topic of digital and online access to a specific group of women suffering from Gynecologic diseases.
In my opinion, the authors should revise the manuscript to improve its readability. As it stands, I do not believe it is ready for publication.
It requires significant revisions in both the methods and results sections.
Addressing these issues would be advantageous for the manuscript's quality.
IntroductionI believe that the Authors could integrate the introduction about the importance of digitalisation or telemedicine also in the case of a disaster such as an earthquake or a pandemic (see for example doi: 10.1007/s12553-023-00762-2 ; 10.1016/S0140-6736(23)00564-0; )
The Materials and Methods section should provide a detailed description of the variables selected for analysis.
The results are not easily read due to the variables reported in Table 1, in Figures, and in the text of the manuscript.
For example:
Table 1 reported the “age” as a continuous variable and the Authors reported data for age >= 50y and age >=60; honestly, I don’t understand why.
Figure 1 reported age as a categorical variable and in the text the analysis dichotomous variable while in Figure 1 it was indicated as a categorical variable …. I suggest indicating how the variable “age” was analysed
Figure 1 is a very poor indicator of the results; I suggest reporting all results using tables with n and percentage or mean and standard deviations and the test carried out with the p-value
The statistical analysis section should include the variables that were investigated and analysed, based on the statistical test used and the statistical method used to analyse and report the results of the computed Digitalization index.
All figures are not clear. If you are using the boxplot, it would be more beneficial for the authors to report data trends and variability (median and interquartile differences) differently.
Reviewer 2 Report
Comments and Suggestions for Authors
The study was carried out by selecting 150 patients from the database of followed patients. There was hesitation regarding the fact that it was a cohort study. It is more appropriate to call it a cross-sectional study.
What criteria were used to determine the sample size? , The power of the study was taken as the number of Type I, and Type II errors.
The total numbers given in Table 1 contradict each other. It does not total 150.
In Table 1, the age distribution should be <49/50-59/;60-69;/>70.
Distribution as >60 and >70 is not appropriate.
In general, digitalization in disease groups is increasing as a result of technological development why gynecological cancers are taken as a basis.
What is the internal consistency coefficient of the scale used? (Cronbach alfa? KR20 ?)
The results given between lines 133-149 should be presented in tabular form.
Figure 2A should be removed. It is a difficult graph to understand.
As a result, "The predominant majority of gynecologic cancer patients can be reached by modern internet-based eHealth technologies, yet the digitalization status of their patients each patient should be obtained at the beginning of therapy to optimize digital patient-physician interaction and eHealth offers." has been given. The conclusion sentence and the expected result of the research do not match well and remain very simple.
In the cohort study; Giving the Relative Risk, Attributed Risk, and Protectability Speed in the digitalized group is expected to reduce the risk. No result has been given regarding these.
Round 2
Reviewer 1 Report
Comments and Suggestions for Authors
The Authors have reported all suggestions. I believe that the manuscript is ready to be published.
Reviewer 2 Report
Comments and Suggestions for Authors
Corrections made by the author are sufficient. Acceptable.
Capital P ( (P=0.024), educational background (P=0.013), the availability of a computer (P<0.001), own given in the text computer experience (P<0.001) and the availability of internet at home (P<0.001), yet not with cancer entity, household size or origin of the patient. ) values should be corrected to lowercase p.
